# Test Bench for Highly Segmented GRIT Double-Sided Silicon Strip Detectors: A Detector Quality Control Protocol

**DOI:** 10.3390/s23125384

**Published:** 2023-06-07

**Authors:** J. A. Dueñas, A. Cobo, L. López, F. Galtarossa, A. Goasduff, D. Mengoni, A. M. Sánchez-Benítez

**Affiliations:** 1Departamento de Ingeniería Eléctrica y Centro de Estudios Avanzados en Física, Matemáticas y Computación, Universidad de Huelva, 21007 Huelva, Spain; 2Dipartimento di Fisica Astronomia dell’ Università di Padova, 35131 Padova, Italy; alex.cobozarzuelo@studenti.unipd.it (A.C.); daniele.mengoni@unipd.it (D.M.); 3Dipartimento di Elettronica Informazione e Bioingegneria dell’ Politecnico di Milano, 20133 Milano, Italy; luis.lopez@mail.polimi.it; 4INFN, Sezione di Padova, 35131 Padova, Italy; franco.galtarossa@lnl.infn.it; 5INFN Laboratori Nazionali di Legnaro, 35131 Padova, Italy; alain.goasduff@lnl.infn.it; 6Departamento de Ciencias Integradas y Centro de Estudios Avanzados en Física, Matemáticas y Computación, Universidad de Huelva, 21071 Huelva, Spain; angel.sanchez@dfaie.uhu.es

**Keywords:** double-sided silicon strip detector, silicon detector tech bench, GRIT collaboration

## Abstract

This work deals with the characteristics of highly segmented double-sided silicon detectors. These are fundamental parts in many new state-of-the-art particle detection systems, and therefore they must perform optimally. We propose a test bench that can handle 256 electronic channels with off-the-shelf equipment, as well as a detector quality control protocol to ensure that the detectors meet the requirements. Detectors with a large number of strips bring new technological challenges and issues that need to be carefully monitored and understood. One of the standard 500 μm thick detectors of the GRIT array was investigated, undergoing studies that revealed its IV curve, charge collection efficiency, and energy resolution. From the data obtained, we calculated, among other things, the depletion voltage (110 V), the resistivity of the bulk material (9 kΩ·cm), and the electronic noise contribution (8 keV). We present, for the first time, a methodology called “the energy triangle’’ to visualize the effect of charge sharing between two adjacent strips and to study the hit distribution with the interstrip-to-strip hit ratio (ISR).

## 1. Introduction

Large particle detector systems are fundamental tools in understanding the properties of the atomic nucleus; in particular, those nuclei that are away from the valley of stability. This quest is one of the most challenging and effort-demanding in nuclear physics. The European GRIT (granularity, resolution, identification, and transparency) project [1] aims to perform studies of shell structure and shape evolution, nuclear pairing, transfer reaction, etc., away from stability. The GRIT system is a 4π silicon detector array that is fully integrable in other systems such as AGATA [2] and PARIS [3]. The array of detectors consists of a number of stacked (two or three units) double-sided silicon strip detectors (DSSSD), which will provide more than 7000 electronic channels of information. For more details about GRIT, the reader is referred to [4,5]. Most of the success in reaching new knowledge will rely on the detector’s capability for discriminating between the different masses of the impinging particles. For this purpose, many works have been carried out during the last 20 years, such as [6,7,8,9]. It is important, therefore, to ensure that the detectors delivered by the manufacturers meet the required standards. To this end, the detectors must be tested before the final assembly in GRIT. In this work, we present a dedicated test bench for highly segmented DSSSD that is able to handle 256 electronic channels, employing off-the-shelf equipment. With it, we also propose a novel detector quality control protocol, to help to not only unify criteria among the different labs working within GRIT, but also to economize on equipment and time. As an example of this working principle, one of the GRIT DSSSDs was placed on the tech bench, and we could study its IV curve to obtain its depletion voltage, in order to calculate its energy resolution and also to look into the particles that impinged on the SiO${}_{2}$ interstrip areas. All this information revealed the quality of the detector under study. This paper is organized as follows: Section 2 describes both the experimental framework (i.e., detector and electronic signal processing) and the detector quality control protocol. Section 3 then combines the results and evaluates the quality of the detector. Finally, Section 4 presents our conclusions.

## 2. Materials and Methods

This section deals with the main blocks of the test bench, which are shown in schematic form in Figure 1. The fact of having a minimum number of electronic blocks on the test bench not only hugely reduces the electronic noise contribution but also make the system portable and economically viable. Briefly, the detector under test (DUT) was kept in a vacuum chamber ($10^{-5}$ mbar during data collection). With the use of an interface circuit (PCB-adaptor), the detector was connected to the cables that take the signals out of the chamber. These signals were fed into the preamplifiers and finally digitized for analysis. The DUT, the PCB-adapters, and the electronic chain are described in the following subsections. We proposed an acceptance–rejection detector test protocol, with the aim of providing the best quality detectors for our experiments.

### 2.1. Double-Sided Silicon 128 × 128 Strip Detector

The GRIT charged particle array is a compact, high granularity, 4π acceptance silicon detector array. The design of GRIT is based on a conical-shaped set of eight trapezoidal telescopes in both the forward and backward hemisphere with respect to the beam direction, assembled with a ring of squared-shaped silicon telescopes around $90^{\circ }$, achieving a 4π solid angle. Most of these detectors are made out of neutron transmutation doped silicon $6^{\prime \prime }$ wafers, cut at 5 degrees off axis and reverse mounted, to improve the particle identification by means of pulse shape analysis. Several detectors have been delivered by Micron Semiconductor Ltd. (Lancing BN15 8SJ, UK) [10] to the GRIT collaboration, from which we selected one with trapezoidal geometry, since the length of the strips changes across the detector active area; its characteristics are shown in Table 1. The DUT was fabricated on a 500μm thick NTD silicon wafer. On the ohmic side, each N${}^{+}$ electrode is surrounded by a P${}^{+}$ implantation (P-stop), in order to prevent the formation of conductive paths between adjacent electrodes. The manufacturer claimed (i) a full depletion voltage of >100 V, (ii) a strip leakage current <20 nA (<100 nA for end-strips), (iii) a total leakage current <2 μA at full depletion voltage plus 30 V, and (iv) an energy resolution ≤40 keV. These are within the technical requirements i.e., operating bias voltage ≤200 V, total leakage current at bias voltage <2 μA, and energy resolution ≤40 keV.

### 2.2. PCB-Adaptor Design

To be able to process the signals coming from the detector, it is necessary first to take them out of the vacuum chamber and send them to the preamplifiers. Due to the micro-dimensions of the molex connectors embedded in the kapton cables of the detector and the connector employed by the preamplifiers, it was necessary to create a vacuum-resistant interface that would allow us to transfer the signal from the detector to the preamplifiers. Therefore, it was one of our objectives to design and manufacture this electrical interface or PCB-adaptor, bearing in mind the pin assignment of the Mesytec preamplifier MPR-64. Taking into account the length of the detector’s kapton flex cables and the distance between the embedded molex connectors, it was decided to manufacture one PCB-adaptor per molex connector i.e., each PCB-adaptor will read 64 strips, which will help to reduce the stress that can build up along the margin of the kapton flex cable pulling the cables connected to the feedthroughs. Taking into account IEEE design recommendations for electronic circuits, ninety degree angles were avoided when tracing the tracks. A ground plane was used for ground (GND) signals, and the maximum current that the track can support (when the thickness and width are 1 oz/ft${}^{2}$ and 6 mil respectively) is 0.61 A. To complete this task, Eagle software for PCB design was employed. The design was carried out in the “Laboratorio de Interacciones Fundamentales” LIFE (Spanish for Laboratory of Fundamental Iteractions) of the University of Huelva, which is a member of the GRIT collaboration. Figure 2-left shows the layout of the tracks, as given by the software, and a picture of the final product is also shown in Figure 2, right. It can be seen that each PCB-adaptor has a mating molex connector and 4 male pin connectors of 20 pins each (16 signals + 4 pins for other purposes), giving a total of 64 signals.

### 2.3. Electronic Signal Processing

Two stages of signal processing are present, the analog and the digital stages. For both stages, off-the-shelf equipment was purchased, which means that this test bench could easily be assembled at any of the collaborating laboratories. The analog end of the electronic chain consists of four multichannel charge-sensitive preamplifier modules specially designed for double-sided multi-strip silicon detectors, each of them handling 64 channels. The manufacturer of the preamplifier model MPR-64 is Mesytec GmbH & Co. (85640 Putzbrunn, Germany) [11]. A summary of the features of these preamplifiers is shown in Table 2. The two MPR-64 modules processing the 128 P-strip signals had an energy range of 25 MeV, while the two modules for the 128 N-strip signals had an energy range of 20 MeV, this was not intentional but these were the modules available.

A pulser was distributed to all the modules, so that each electronic channel’s constraint in terms of, for example, energy resolution could be ascertained. In the meantime, a positive bias voltage was applied to the bias input of the MPR-64 modules reading the N-strips (reverse bias), while the bias inputs of the rest of the modules were sent to ground through a 50Ω Lemo terminator. More details about how the bias voltage is applied to the detector are given in the IV curve subsection. The preamplifier outputs are realized as differential outputs for twisted pair 34-pin male header connectors with an amplitude ranging between 0 and ±1 V. The standard decay time is about 25μs, with a rise time of 12 ns for a 0 pF input capacity.

Finally, the analog signals are digitized by a the CAEN V2740 digitizer [12], whose main features are shown in Table 2. It should be mentioned that the preamplifier output signals are connected to the digitizer through a 64-channel 2.54 mm male header connector adapter (CAEN A372F). A user-friendly interface designed by CAEN called CoMPASS was employed to control the digitizer and perform basic mathematical analysis of the recorded data. A digital trapezoidal filter was used to obtain the energy spectrum, with parameters set to 3μs rise time, 1.6μs flat top, and 22μs pole zero. Two digitizers were employed, meaning that only 128 detector signals could be recorded at the same time. It was also decided to only save the energy values (no waveforms), given as the output values of the trapezoidal filter in a ROOT tree format file.

### 2.4. Detector Quality Control

It was of paramount importance to ascertain whether or not the delivered detector units met the requirements. In order to do that, the detector units had to undergo a number of tests to make sure that they are suitable to be installed in GRIT. Figure 3 shows a workflow diagram that aimed to characterize the DUT and help to make a decision on whether the DUT can be or cannot be accepted.

The first stage of the detector quality control (DQC) is visual inspection with the naked eye, in order to detect any defects (scratches or stains) on the surface of the detector’s active areas. Contamination of the surface could create leakage paths. Then, employing a magnify glass or a similar device, one must make sure that the bondings are secured and that there is no other damage around the silicon material, as mechanical damage at the edge leads to very large leakage of current.

If no problems have been spotted, in the second stage of the DQC, the DUT should be connected to the setup shown in Figure 1. Next, the total IV curve should be obtained by recording the current flowing for different applied voltages. The term “total IV curve” refers to the fact that only one bias supply unit is employed to polarize the whole detector, and therefore the recorded current is the sum of the leakage current of all detector strips. It is important not to surpass either the current or voltage limits established by the manufacturer. This procedure can be performed manually, recording directly from the bias supply unit the values indicated in its display or employing an IV tracer i.e., test equipment that sweeps an electrical load and measures both the current and voltage at multiple points during the sweep. The leakage current can be a “whistleblower” that helps raise the alarm about, for instance, bulk damage and radiation damage (exploits in dosimetry). Increased current leakage may cause (i) the integrated current over typical signal processing times to greatly exceed the signal, (ii) the shot noise to increase, and (iii) the power dissipated in the detector to increase (bias current times voltage). After these two tests, a decision can be made to carry on with the DQC or reject the DUT.

The third stage assesses the fraction of the total charge produced during an ionization event that is collected in the detector strip electrode for later readout i.e., the charge collection efficiency (CCE), and also how fast the charge carriers (i.e., electrons & holes) are swept from the depletion region by the applied electric field. For a fully depleted detector, with a large electric field, CCE is approximately 100% for the full active volume. In regions of the detector with a lower electric field, the CCE could be less than 100%, due to charge recombination. A radiation source is used to generate the electrical signal that will provide us with the CCE curves. We will record how the main peak centroid shifts (from one ADC channel to another) as a function of the applied bias voltage from its digital energy spectrum. The y-axis variable (i.e., ADC channel) will increase linearly with the applied voltage (x-axis variable) until it reaches a plateau, i.e., the applied voltage no longer has effect. The experimental depletion voltage $V_{d}$, i.e., the voltage where the depletion width equals the thickness of the bulk d=0.0497 cm, can be obtained from this curve; in turn, the other parameters can be estimated, such as the dopants concentration or the resistivity of the material. To calculate the dopant concentration in the bulk of an n-type material (N), this well-known formula [13,14,15] can be applied:(1)$$V_{d}=\frac{N\;e\;d^{2}}{2\;\epsilon_{Si}\;\epsilon_{0}}$$
where $e=1.6\times 10^{-19}\;C$ is the electronic charge, $\epsilon_{Si}=11.9$ is the dielectric constant of Si, and $\epsilon_{0}=8.854\times 10^{-14}$ F/cm is the permittivity of vacuum. In addition, the resistivity of an n-type material ($\rho_{n}$) can be calculated [13,14,15] by:(2)$$V_{d}=4\left[\frac{\Omega \cdot \mathrm{cm}}{\mu m^{2}}\right]\frac{d^{2}}{\rho_{n}}-V_{bi}$$
where $V_{bi}$ is the “built-in” junction potential, which is typically about 0.5 V. The thickness of the bulk should be given in μm i.e., d=497μm, so the resistivity $\rho_{n}$ is expressed in units of Ω· cm. This information will reveal the quality of the manufacturing, and therefore help with the decision-making process.

The fourth and last stage of the DQC relies on the energy distribution of the incident radiation, i.e., energy resolution. Nevertheless, electronic noise is still important in determining the minimum detectable signal, i.e., the detection efficiency. Again, to proceed with the energy resolution test, an alpha source is needed. This time, the detector’s bias voltage is usually set above the depletion voltage (e.g. $V_{d}+20$ or +30 V), and the spectrum of the alpha source is recorded for both P- and N-strips. For our test bench, the analog charge signals coming out from the preamplifiers are digitized before applying any signal processing. Of course, since all amplifiers have a limited bandwidth, every amplifier is a pulse shaper. The final energy resolution is estimated using software i.e., each digital signal is shaped by a trapezoidal filter with optimized shaping parameters: rise time, flat top, and decay time constant, with the amplitude plotted as a histogram or spectrum. The full width at half maximum (FWHM) is extracted from a Gaussian fit of the main peak. A semiempirical function for the obtained energy resolution ($W_{FWHM}$) is given as an addition in quadrature of electronic noise ($W_{N}$) and the FWHM for a single α-particle contributions:(3)$$W_{FWHM}=\sqrt{W_{N}^{2}+W_{S}^{2}+W_{P}^{2}+W_{C}^{2}}$$
where $W_{S}$ is the contribution from energy straggling, $W_{P}$ from the carrier production, and $W_{C}$ from the carrier collection. We used LISE${}^{++}$ to calculate the stopping and range of ions in matter, assuming 1μm Silicon equivalent as the dead layer thickness, giving an expected $W_{S}$ value of about <20 keV. In addition, considering $W_{P}=2.35\sqrt{FE_{Si}E_{\alpha }}$, being F=0.1 the Fano factor, $E_{Si}=3.6$ eV, the energy required to produce an electron–hole pair, and $E_{\alpha }$ the alpha particle incident energy [13,14,15]. The expected value of $W_{P}$ is therefore 3.4 keV for 5.805 MeV alpha (${}^{244}Cm$). All in all, it is expected that the total α-particle contribution is mainly due to the energy straggling ($W_{S}$) in the dead layers, since $W_{C}$ should be negligible for a over-depletion bias voltage. The last step in the DQC process is the comparison between the obtained $W_{\alpha }$, the one given by the manufacturer, and the one given as a requirement. This will provide the information necessary to decide whether to accept or reject the DUT.

In summary, the DQC protocol has two main decisions points: the earlier one involves the first two stages, which can be performed in a reasonably short time. For the second decision point, a longer time is needed to gather sufficient statistics and digitally analyze the recorded data. It is our estimation that two 128×128 DUT can undergo the DQC protocol within a working day, assuming all the equipment runs smoothly.

## 3. Results and Discussions

The testing workbench was first assembled at the Laboratori Nazionali di Legnaro (Legnaro, Italy). The results obtained from the DUT are shown and discussed in this section, following the DQC protocol presented above. To help the readers understand the results and the discussions below, we show a photograph of the detector setup in the vacuum chamber in Figure 4.

### 3.1. IV Curve

The different blocks or equipment used in the generation of the IV curve are depicted in Figure 5. The bias voltage source that we used is connected to the preamplifier (MPR-64) detector bias input, while the detector itself is connected to the preamplifier. The MPR-64 has four input stages (SubD25 female connector), and each stage has its own bias input that deals with 16 channels or strips at the same time. The $V_{Bias}$ is applied through a RCR network to each channel stage. The biasing resistors must be large enough to reduce electronic noise, and the bypass capacitor *C* shunts any external interference coming through the bias supply line to ground. It should be considered then that there will be a small voltage drop due to the biasing resistors, e.g., for a 100 nA current and 10 MΩ resistors, the drop would be 1 V.

Figure 6 shows the values obtained for the total leakage current for a voltage range from 0 to 250 V. The values given by the manufacturer are also shown. Apart from the MHV-4 bias supply unit, a Keithley 2450 source measure unit (SMU-2450) was also employed to double check the readings given by the MHV-4 display, although the SMU-2450 was restricted to ±200 V. Three main points can be taken from the IV curves that were generated: (i) the differences between the leakage current values obtained and those given by the manufacturer were threefold, (ii) no significant difference could be seen when using any of the two bias sources, and (iii) the “hump” feature above 100 V. The manufacturer gave no information about how it measured the leakage current. However, the value given by the manufacturer of a total leakage current <2μA for a over-bias/depleted voltage of Vd+30 V was met. The obtained $V_{d}$ will be discussed later. The manufacturer recommended bias was 140 V, as depicted in the IV curve, Figure 6.

Next, we give a plausible explanation for the sudden jump of the leakage current value close to the depletion voltage. In silicon detectors, the leakage current is affected by the generation of minority carriers in the depletion region. In many cases, the depletion depths extend to hundreds of microns. These minority carriers can also arise from the undepleted region and from the surface. If a significant amount of “impurities” are still present after the fabrication process, the leakage current increases. To avoid this happening, there are the so-called called gettering techniques [16], which basically rely on the formation of an impurity “sink” layer on the back side of the wafer [17]. It must not be forgotten that the detector encapsulation and the guard rings can also contribute to the rise of the leakage current. Further studies are planned, in order to achieve a detailed understanding of the leakage current behavior. The fact that the DUT exhibited this abnormal IV response does not necessarily mean that it is not suitable for our purposes; therefore, it was moved forward to the next stage of the DQC protocol.

### 3.2. Charge Collection Efficiency

Figure 7 shows the shifts of the three Gaussian-fitted main peaks (i.e., ${}^{239}$Pu, ${}^{241}$Am, and ${}^{244}$Cm) of an α-source as a function of the applied bias voltage. The spectra were obtained from four middle P-strips (solid lines) and four N-strips (dashed lines). The difference in ADC channels between the P- and N-strips (i.e., N-strips occupy higher channels) was due to the preamplifier’s sensitivity. Those reading the P-strips had a slightly different sensitivity than for the N-strips. The particles were impinging on the junction side (P-strips), where the electric field had its highest values. Alpha particles in the range of 5.5 MeV stopped in the first 25μm of the Si bulk, i.e., a short path length for holes and long path for electrons, which means that most of the carriers easily and quickly reached the P-strips (electrode) with a small bias voltage applied, as is shown by the obtained P-strip curves (solid line in Figure 7). However, in order to collect the carriers (electrons) on the opposite electrodes (N-strips) we needed to extend the depletion region to the full width of the bulk (i.e., detector thickness), and that implied a significant increase in the bias voltage. The signal amplitude distribution was quite broad at low voltages and was also linearly related (dashed lines N-strips Figure 7). The depletion bias of the detector was then marked by the bias voltage above which the amplitude of the signal was no longer dependent on the electric field, i.e., there was no change in the number of ADC channels. The DUT showed a depletion voltage of about $V_{d}\approx 110$ V.

Having experimentally obtained the $V_{d}$, the resistivity of the bulk could be calculated by employing Equation (Equation 2), yielding a value of $\rho_{n}\approx 9$ kΩ·cm, and therefore within the resistivity range given by the manufacturer (Table 1). Similarly, from Equation (Equation 1) the dopant concentration can be extracted to yield a value of $N\approx 5.9\times 10^{11}$ cm${}^{-3}$; this parameter is not given by the manufacturer, but the obtained concentration value is within the typical range for NTD silicon semiconductors, i.e., $10^{11}$-to-$10^{13}$ cm${}^{-3}$ [18]. This agrees with the manufacturer’s recommendation that a reasonable detector bias voltage is 140 V, since $V_{Bias}=V_{d}+30=140$ V. The detector’s total leakage current at 140 V was about 1.05μA, which means that the expected leakage current per strip would be under 10 nA, roughly speaking. This is also in agreement with the information given by the manufacturer.

### 3.3. Energy Resolution

Figure 8 shows the energy resolution (FWHM) obtained for the P- and N-strips when a bias voltage of 140 V was applied. We also show the electronic noise contribution by means of a pulser. In general, most of the strips yielded energy resolutions below 30 keV, although some higher values were given by the strips at the edges of the detector. Far below these values, the energy resolution was limited by the noise contribution of the electronic chain (the electromagnetic environment may change from one lab to another), which was quite constant for most of the preamplifier channels, giving a value of about 8 keV. Nevertheless, the obtained spectra showed the “left-shoulder” of the main peaks quite clearly (Figure 8 left pad), which was a good indicator at first sight. Table 3 summarizes the average and standard deviation values of the energy resolution for the three main α-peaks and the resolution of the injected pulser. These values were calculated from the 128 strips.

If we now turn to Equation (Equation 3), the contribution from the alpha particle to the obtained 26.4 keV total resolution was about $\sqrt{26.4^{2}-8.06^{2}}\approx 25.2$ keV. Therefore, the carrier collection contribution was not negligible, $W_{C}=\sqrt{25.2^{2}-20^{2}-3.4^{2}}\approx 15$ keV. This could be evidence for the presence of charge collection losses due to impurities, which could increase the leakage current, as discussed in the IV curve section. Further studies and new measurements are planned, in order to achieve a detailed understanding of the different contributions to the energy resolution in a setup such as that employed in this paper. Nevertheless, the obtained energy resolution values are adequate for most standard experiments in nuclear physics.

### 3.4. Interstrip Study

In large highly-segmented silicon detectors, the areas covered by the strip electrodes and the SiO${}_{2}$ interstrip insulator may be of the order of 90% and 10% of the total active area, respectively. Therefore, a considerable number of particles will enter the bulk material through the interstrips. Figure 9 shows the electrode structure for the ohmic side of our DUT, where two insulating p${}^{+}$ implants (p-stops) can be seen between the n${}^{+}$ electrodes. Figure 10 shows the energy correlations between two adjacent strips (strip 62 vs. 63) and two non-adjacent strips (60 vs. 62). This energy matrix was formed by the intersection of the two spectra of the triple-α-source. When a particle impinges on a strip electrode, i.e., the $\alpha_{1}$ scenario in Figure 9, the attracted carriers, which were generated during the ionization process underneath it, will create an electrical signal. This only occurs for that electrode or strip and nowhere else. In that case, if we correlate the spectrum of two nonadjacent strips we end up with the intersection of the two spectra, as shown in Figure 10, left pad. However, for particles entering the interstrip SiO${}_{2}$ area, the generated carriers will be subjected to the influence of the two adjacent strips’ electric field, and therefore they will be collected by the two nearest strips, i.e., one particle will generate two electrical signals. The degree of the amplitude is related to the closeness of the particle and the strip; the closer the particle is to the strip, the higher the amplitude of the electrical signal generated on the strip. For a particle “hit” right in the middle of an interstrip area, i.e., $\alpha_{2}$ scenario in Figure 9, each of the adjacent strips will receive half of the total generated carriers i.e., half of the total energy of the particle. As the hits shift towards one of the strips, the total energy of the particle will be unequally distributed between the adjacent strips. This is the scenario depicted in Figure 10, right pad, and this is usually referred as “charge sharing” [19,20,21,22]. It can be seen that the interstrip events are the cause of the hypotenuse side of a right angled triangle, which we will call “the energy-triangle”. We believe that it is worth investigating the possibility that the energy-triangle, as defined in this paper, may be used as a tool for assessing the quality of the interstrip structure.

### 3.5. Hit Distribution

Based on the statistics gathered by the acquisition system for all of the detector’s strips, we can also study the hit distribution generated by the impinging particles. Figure 11 left and middle pads show the bell-shaped distribution obtained for the total and interstrip hit distributions. Judging from the P-strip hit distribution, we can conclude that the α-source was centered on the junction side (P-strips). However, the N-strip distribution was shifted to lower strip numbers. If we had found any deviations from this distribution, this would have alerted us to possible strip damage. It is also advisable to check for the interstrip hit distribution i.e., only hits on the SiO${}_{2}$ areas, which should also follow the same distribution, as shown in the middle pad. To obtain the interstrip distribution, we only counted the events on the hypotenuse of the energy-triangle. Furthermore, the interstrip-to-strip hit ratio (ISR) distribution can also be obtained by counting the events on one of the energy-triangle sides. The ISR distribution presents a U-shaped valley profile, as shown in Figure 11, right pad, which was due to both the relative size of the α-source and the strip position above the detector, i.e., the solid angle of the source. For the irradiated area just in front of the particle source (strip 40 to 90), where the incident angle was basically zero (valley bottom), the ISR was about 3%, i.e., 100 hits on the strip and only 3 on its interstrip. However, as the incident angle of the particles increased so did the ISR, reaching its maxima (nearly 5%) at the edges of the detector, i.e., about 5 hits on the interstrip and 100 on the strip. This, we believe, was caused by the shadow effect that the SiO${}_{2}$ areas (they rose 1μm above the Al electrodes) may have over their neighboring electrodes at large incident angles.

## 4. Conclusions

We have presented a test bench for highly segmented DSSSD that is able to handle 256 electronic channels employing off-the-shelf equipment. Moreover, we developed a detector quality control protocol that uses a new methodology for visualizing the signals generated from particles entering the SiO${}_{2}$ interstrip areas. Two new concepts or tools were presented in this paper: the energy-triangle and the interstrip-to-strip hit ratio distribution. The detector under test yielded an overall energy resolution of 27 keV for the P-strips and 26 keV for the N-strips when the bias voltage was set to 30 V above the depletion voltage, i.e., $V_{d}+30=140$ V. In general terms, the detector met the requirements. However, the IV curve showed an unusual hump-like shape above the depletion voltage, which was attributed to impurities within the bulk material. The proposed test bench can be easily set up in any of the partner laboratories of the European GRIT project. The limitations of our test bench include (i) the detector’s dimensions and (ii) connector types, (iii) the number of electronic channels to be digitized, and (iv) the intrinsic electronic constrains, e.g., equipment noise, minimum detectable signal, etc. Future works will consider a mechanical support structure to hold the detector vertically, so both the junction and ohmic sides can be irradiated. Further studies are planned, in order to achieve a detailed understanding of the different contributions to the energy resolution and the behavior of the IV curve after the depletion voltage.

## Figures and Tables

**Figure 1 sensors-23-05384-f001:**
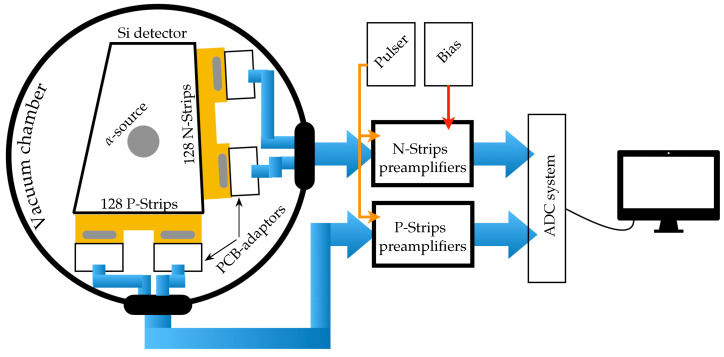
Test bench block diagram. The detector signals are brought to the feedthrough cables via PCB-adaptors. Outside, the preamplifiers are connected to a pulser and a voltage source (Bias). The preamplifier outputs are sent to the ADCs.

**Figure 2 sensors-23-05384-f002:**
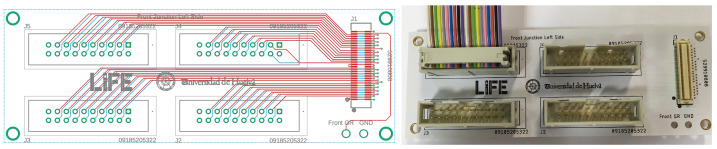
Design of the PCB-adaptor. Example of the layout of the traces (**left**) and photo of the manufactured prototype (**right**). Each PCB-adaptor can handle 64 strips.

**Figure 3 sensors-23-05384-f003:**
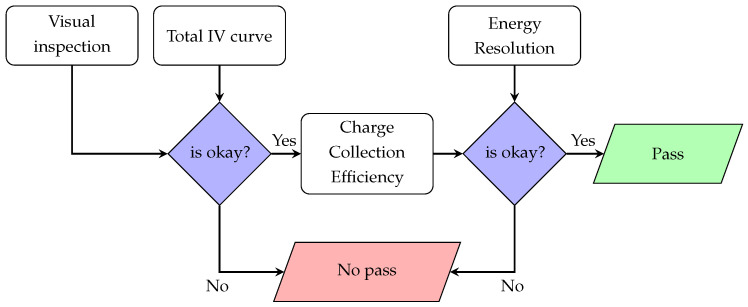
Workflow diagram for the detector quality control test. Four stages or tests are needed to make sure that the DUT meets the requirements and is accepted.

**Figure 4 sensors-23-05384-f004:**
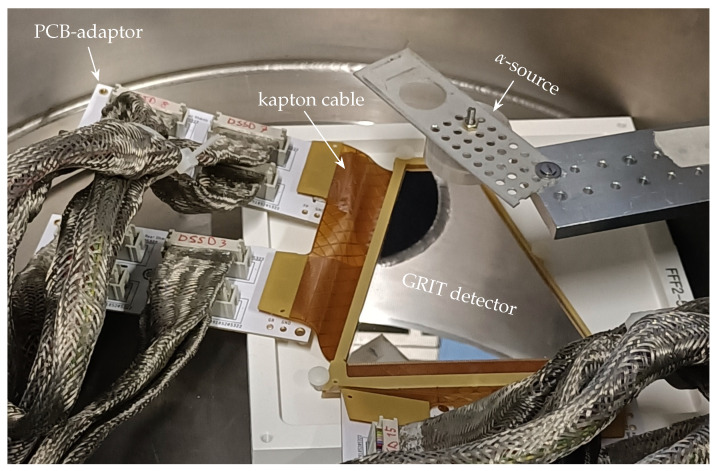
Photograph of the detector in the chamber. The α-source capsule was 6 cm above the detector. The kapton cables’ terminals hold the micro-molex connectors that transfer the signals to the PCB-adaptors and then to the outside.

**Figure 5 sensors-23-05384-f005:**
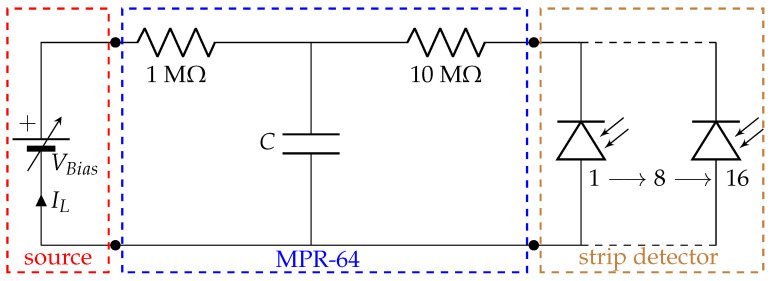
IV curve circuit with the different blocks: the bias source MHV-4, the preamplfier MPR-64 bias RCR network, and the strip detector. For each $V_{Bias}$, there will be a $I_{L}$.

**Figure 6 sensors-23-05384-f006:**
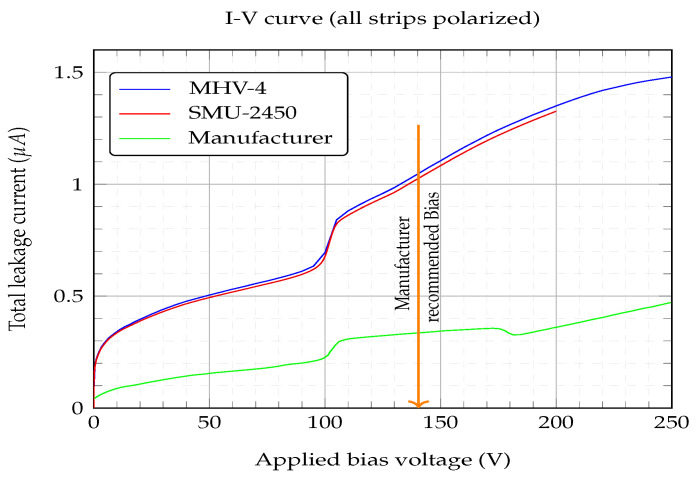
Total leakage current (i.e., all strips polarized) with detector in a $10^{-6}$ mbar vacuum. The bottom curve is the one provided by the manufacturer, Micron Ltd. The manufacturer recommended bias at 140 V is also indicated.

**Figure 7 sensors-23-05384-f007:**
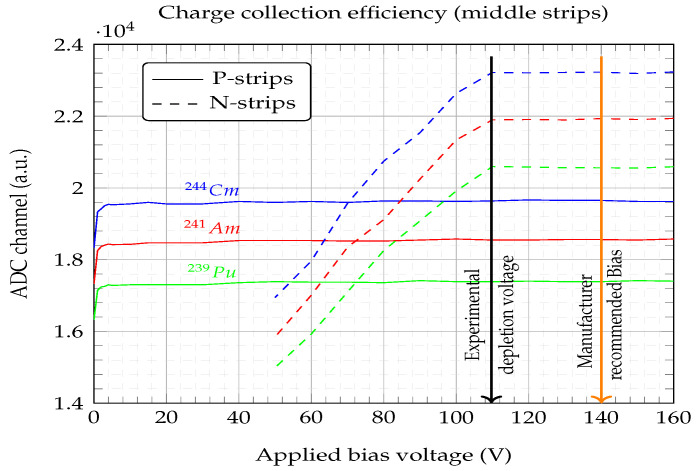
Charge collection efficiency for the three main peaks of the α-source. The depletion voltage was obtained from the N-strip ADC channels at 110 V. The manufacturer recommended bias at 140 V is also indicated.

**Figure 8 sensors-23-05384-f008:**
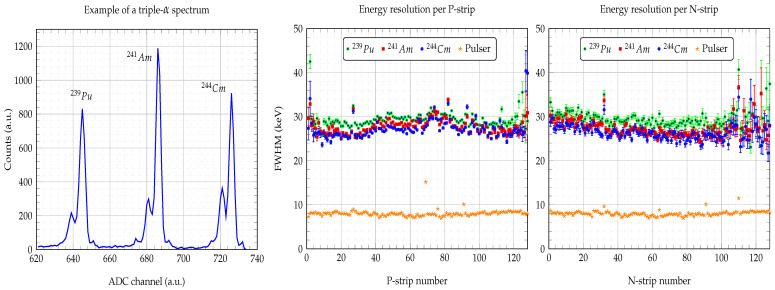
Energy resolution per strip. Example of a triple-α spectrum from one of the strips (**left pad**). Full width at half maximum (FWHM) of each P-strip (**central pad**) and N-strip (**right pad**), for a bias voltage of 140 V. Alpha particles entering through junction side (P-strips). The electronic chain resolution given by a pulser. The FWHM average values for ${}^{244}Cm$ were 27.3 keV for P-strips, 26.4 keV for N-strips, and 8 keV for the pulser.

**Figure 9 sensors-23-05384-f009:**
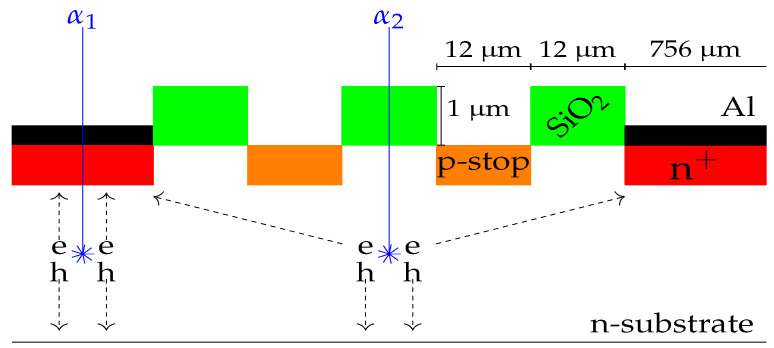
Electrode structure (not to scale) for the ohmic side of the detector. The two insulating p${}^{+}$ implants (p-stops) can be seen between the n${}^{+}$ electrodes. Two scenarios are depicted: $\alpha_{1}$ impinging on one electrode, and $\alpha_{2}$ on the interstrip gap. The produced mobile charge carriers electron–hole (e-h) pairs are depicted.

**Figure 10 sensors-23-05384-f010:**
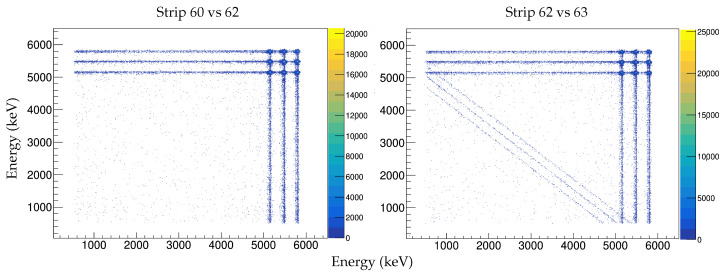
Energy correlation matrix between two strips. The charge sharing effect between two adjacent strips create the energy-triangle, (**right pad**). In the case of two nonadjacent strips the matrix does not show the energy-triangle, (**left pad**).

**Figure 11 sensors-23-05384-f011:**
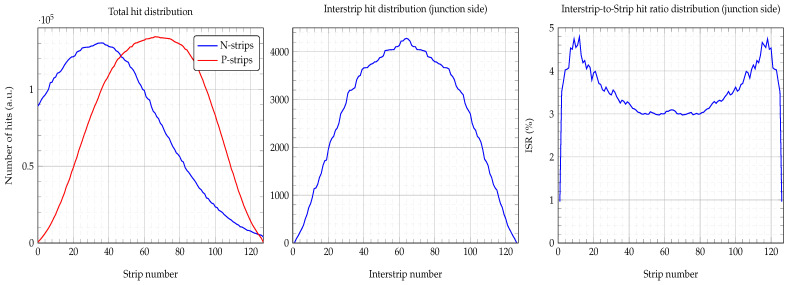
Hit distributions for both P- and N-strips, alpha source placed about the center of P-strips but not at the center of the N-strips, left pad. Interstrip hit distribution for particles impinging on the junction side, middle pad. interstrip-to-strip hit ratio (ISR), right pad.

**Table 1 sensors-23-05384-t001:** Main detector characteristics as given by Micron Semiconductor Ltd. [10].

Parameter	Description
Wafer type	NTD silicon 〈100〉, size $6^{\prime \prime }$, No.: 3447-12
Substrate type	N-Type Silicon (NTD 5-degree off axis)
Resistivity	7–10 kΩ·cm
Thickness	497μm, total thickness variation of ±1μm
Detector type	Double-sided ion implanted totally depleted structure
Implantation	Boron for junction side (P-strips) and phosphorus for ohmic side (N-strips)
Strip no.	128 P-strips and 128 N-strips i.e., 128×128
Strip pitch	715μm for P-strips and 816μm for N-strips
Strip separation	60μm for both P- and N-strips
Isolation	2 P-stop structure between N-strips
Metallizing	Aluminum 3000 Å
Dead layer	<1μm
PCB	trapezoid minor 32 mm, major 110.4 mm, side 123.7 mm, thickness 5.6 mm
Outputs	via Molex 53916-0808 embedded in Kapton flex cable

**Table 2 sensors-23-05384-t002:** Preamplifier and digitizer main features, as given by Mesytec [11] and CAEN [12].

Parameter Preamp	Description
Preamplifier model	Mesytec MPR-64
Input stage	4 SubD25 female connectors (16 channels each)
Output stage	Differential output for twisted pair 34 pin male header connector
Output amplitude	0 to ±1 V
Energy range	5–25 MeV for P-strips and 20–100 MeV for N-strips
Bias input	Lemo connector, maximum voltage ±400 V
Pulser input	internally distributed to individual charge termination capacities
**Parameter Digitizer**	**Description**
Digitizer model	CAEN V2740
Analog Input	64 channels 100Ω differential, 50 MHz (−3 dB), ±1.25 V
Digital Conversion	16 bits resol., 125 MS/s simultaneously on each channel
Performance	ENOB: 11.7 (typ.) and RMS: 3.9 LSB

For more details we referred the reader to the manufacturer online Data sheets.

**Table 3 sensors-23-05384-t003:** FWHM average values for P- and N-signals, and for the pulser.

Preamplifier	Pulser	${}^{239}\mathrm{Pu}$	${}^{241}\mathrm{Am}$	${}^{244}\mathrm{Cm}$
P-strips	8±0.8 keV	29.4±1.8 keV	27.7±1.6 keV	27.2±2.3 keV
N-strips	8±0.6 keV	29.9±2.4 keV	27.3±1.9 keV	26.4±1.8 keV

Average and standard deviation out of the 128 strips.

## Data Availability

Not applicable.

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
