# Peer review of "Test Bench for Highly Segmented GRIT Double-Sided Silicon Strip Detectors: A Detector Quality Control Protocol"

_sensors, 2023, doi:10.3390/s23125384_

Round 1

Reviewer 1 Report

The manuscript is well written that it presents a test bench for highly segmented DSSSD and a detector quality control protocol that uses a new methodology for visualizing signals. I have some comments and hope they can be addressed before publication:

1. Is it possible to have some visual illustration for the N and P-strips? For some readers who are out of this field like me, its a bit challenging to picture them and figure out how they work.

2. For the visual check on defects discussed in section 2.4, can you use a figure to show the comparison between pass/no pass samples?

3. Are there any future work to be discussed in the conclusion?

Some sentences need a bit of grammar checking but in general the English writing is fine.

Author Response

The authors would like to thank the reviewer for the feedback received and also to acknowledge the improvement of the new version of the paper due to the reviewer’s comments. Please, find our reply to every point made:

  1. Is it possible to have some visual illustration for the N and P-strips? For some readers who are out of this field like me, it's a bit challenging to picture them and figure out how they work.

Illustrations like these can be found in several of the references given in the paper [13-15 & 19,20]. It is basically a standard thing for Si radiation detectors. 

  1. For the visual check on defects discussed in section 2.4, can you use a figure to show the     comparison between pass/no pass samples?

A new figure (Fig. 4) of the detector in the chamber has been added. Unfortunately, for this work we have only this sample, which presented no problems.

  1. Are there any future work to be discussed in the conclusion?

Future works are now indicated in the conclusion section.

Reviewer 2 Report

The authors present an experimental set-up for testing highly segmented GRIT double-sides silicon strip detectors.

Description of experimental set-up and of the device under test is done in the text with almost no drawing or diagram that could help in understanding the shape and configuration of the detector, the relative position of the alpha sources with respect to the detector and so on. This makes understanding the experimental approach and experimental results quite challenging. 

The interpretation of the experimental results are not at all clear.

For instance, there is no clear interpretation of the "hump" feature, possibly because without a detailed diagram of the devices, it is difficult to attribute meaning to several statements. Moreover, some statements are simply wrong. For instance, the statement ". It must be remembered that impurities are needed to introduce energy levels near the center of the silicon bandgap (doping)" is wrong for two fundamental reasons: 1) introducing energy levels near the center of the silicon bandgap is the last thing one wonts to do in the case of a detector, since these energy levels promote fast charge recombination; 2) doping is the process by which acceptor or donor energy levels are introduced in the bandgap very close to either the valence or the conductance band, for P and N doping, respectively.

In listing the terms appearing in Eq. 1, d is called the "detector thickness" without any other explanation. "d", as used in Eq. 1,  can only be the depletion region extension in a p+/n junction, but it is not clear where this depletion region is supposedly located within the device (as I stated above, no drawing of the device is included).

When dealing with the "Hit distribution" the statement is made that a "Gaussian-like distribution" should be obtained (no reference is given). This is quite surprising, since I do not see any reason for a Gaussian distribution when assuming a point source that emits high energy particles randomly in all directions. It seems to me that the "Hit distribution" should be simply the result of the geometric configuration, may be with some corrections due to "reflection" effects at shallow angles, if any.

Finally, the authors introduce what they call the "energy triangle" which is the result of the observation of the energy correlation of electrical pulses on adjacent electrodes caused by particles physically hitting the spacer between the two. The fact that a straight segment is obtained is somewhat peculiar, but the statement that it can possibly be used to pinpoint anomalies is in no way justified by the authors. The mere observation of a feature does not imply, in itself, that it can be useful in any way.

Author Response

The authors would like to thank the reviewer for the feedback received and also to acknowledge the improvement of the new version of the paper due to the reviewer’s comments. Please, find our reply to every point made:

Description of experimental set-up and of the device under test is done in the text with almost no drawing or diagram that could help in understanding the shape and configuration of the detector, the relative position of the alpha sources with respect to the detector and so on. This makes understanding the experimental approach and experimental results quite challenging. 

The position of the alpha source is now indicated in Fig. 1. A new figure (Fig. 4) of the detector in the chamber has been added. The following paragraph about the detector geometry has been inserted in section 2.1.

“The design of GRIT is based on a conical-shaped set of 8 trapezoidal telescopes in both the forward and backward hemisphere with respect to the beam direction assembled with a ring of squared-shape silicon telescopes around 90°, achieving ∼4π solid angle.” …”from which we have selected one with trapezoidal geometry since the length of the strips changes across the detector active area”

The interpretation of the experimental results are not at all clear. For instance, there is no clear interpretation of the "hump" feature, possibly because without a detailed diagram of the devices, it is difficult to attribute meaning to several statements.

The detector geometry is shown in Fig. 1 and its dimensions are indicated in table 1. Furthermore, we have now added the photograph of the detector in Fig. 4.

Moreover, some statements are simply wrong. For instance, the statement ". It must be remembered that impurities are needed to introduce energy levels near the center of the silicon bandgap (doping)" is wrong for two fundamental reasons: 1) introducing energy levels near the center of the silicon bandgap is the last thing one wants to do in the case of a detector, since these energy levels promote fast charge recombination; 2) doping is the process by which acceptor or donor energy levels are introduced in the bandgap very close to either the valence or the conductance band, for P and N doping, respectively. 

In order to not mislead the readers, we have remove that statement, and also indicate that “Further studies are planned in order to achieve a detailed understanding of it”

In listing the terms appearing in Eq. 1, d is called the "detector thickness" without any other explanation. "d", as used in Eq. 1,  can only be the depletion region extension in a p+/n junction, but it is not clear where this depletion region is supposedly located within the device (as I stated above, no drawing of the device is included).

To clarify this, we have added the following sentence: “The depletion voltage is understood as the voltage where the depletion width equals the thickness of the bulk d”

When dealing with the "Hit distribution" the statement is made that a "Gaussian-like distribution" should be obtained (no reference is given). This is quite surprising, since I do not see any reason for a Gaussian distribution when assuming a point source that emits high energy particles randomly in all directions. It seems to me that the "Hit distribution" should be simply the result of the geometric configuration, maybe with some corrections due to "reflection" effects at shallow angles, if any.

We apologize for misleading the reviewer. The alpha source was embedded in a cylindrical polymeric holder with a circular aperture i.e.  distribution similar to the diffraction by a circular aperture. In the newly added Fig. 4 we show the alpha particle capsule.   

Finally, the authors introduce what they call the "energy triangle" which is the result of the observation of the energy correlation of electrical pulses on adjacent electrodes caused by particles physically hitting the spacer between the two. The fact that a straight segment is obtained is somewhat peculiar, but the statement that it can possibly be used to pinpoint anomalies is in no way justified by the authors. The mere observation of a feature does not imply, in itself, that it can be useful in any way.

The charge sharing effect depends very much on the electric fields (EF) of the adjacent strips, and therefore, any alteration of the EF will alter how the particles reach the strips  e.g. it’s well known that accumulation of charges under the electrodes modified the EF under it. Should one encounter no straight diagonal line, it would mean that charge sharing is uneven between two adjacent strips, which is not the normal behavior of a strip detector. 

Reviewer 3 Report

Dear Authors, 

The manuscript submitted for review deals with the practical aspects of determining detector quality and building a test bench for highly segmented DSSSD. The work has a very high application potential. The issues presented in the manuscript are described in an understandable way.

1. After reading the manuscript, I have a question on the basis of what was the detector with trapezoidal geometry selected?

2. How should formula (3) be interpreted? If the component of equation Wα is the sum of the individual components derived from the various parameters such as WS (a), Wp (b) and Wc (c), then when this sum is squared we should expect a formula like: a2+2ab+2ac+b2+2bc+c2. What is the mathematical justification for the formula (3) used?

3. In the text of the manuscript, there should be references to each formula number.

4. The authors mention in several places the quality control of the detector, understood as making sure that the detector units are free of mechanical defects and impurities (dirt, dust, stains, etc.). Assuming that the test system met the highest quality requirements, the question arises as to what specific contaminants the authors write about in their conclusions...

Sincerely yours, 

Reviewer

Dear Authors, 

I do not find many grammatical errors or mistakes related to the use of incorrect words in the manuscript. The manuscript reads well despite some linguistic inaccuracies. The errors can be divided into several categories, but each of the following comments should primarily be considered as suggestions.

1. For grammatical reasons, you might consider changing individual words: 'system'>'systems', 'brings'>'bring', 'describe'>'describes', 'detectors'>'detector', 'others'>'other', 'N-strips'>'N-stripes', 'decisions'>'decision', 'difference'>'differences', Lines: 2, 5, 37, 98, 166, 186, 212. 

2. You might consider removing some words from the text: 'the energy-triangle'>'energy-triangle', 'in order to', 'or not', 'to make'>'make', 'can be'>'can', 'it' Lines: 11, 58, 115, 118, 119, 315.

3. you can consider replacing individual words with: 'like'>'such as', 'about'>'on', 'rely'>'depend', 'delivered'>'supplied', 'off-the-shell'>'standard', 'labs'>'laboratories', 'put on'>'placed', 'adaptor'>'adapter', 'for completing'>'to complete', 'out in'>'out at', 'analog'>'analogue', 'details about'>'details of', 'can not'>'cannot', 'employing'>'using', 'maginfy'>'magnifying', 'employed'>'used', 'no to surpass'>'not exceed', 'can'>'to', 'to carry on'>'whether continue', 'Also'>'Furthermore', 'The biasing'>'Bias', 'of'>'from', 'threefold'>'three times', 'about'>'on', 'at about'>'to approximately', 'not'>'no', 'out of'>'from', 'in comparison with'>'compared to', 'that'>'which' Lines: 21, 23, 24, 27, 30, 87, 32, 33, 69, 78, 79, 88, 99, 119, 122, 123, 129, 131, 138, 140, 157, 201, 206, 212, 214, 219, 247, 270, 308, 337.

4. In some words, you might consider using a hyphen: ''pre-amplifier'', ''semi-empirical'', ''electro-magnetic'', Lines: 100, 172, 265.

5. certain sentences need re-editing to make them more readable: Lines: 95-96, 104-105, 116-117, 176-177, 256-257, 299-300.

Sincerely yours, 

Reviewer

Author Response

The authors would like to thank the reviewer for the feedback received and also to acknowledge the improvement of the new version of the paper due to the reviewer’s comments. Please, find our reply to every point made:

1. After reading the manuscript, I have a question on the basis of what was the detector with trapezoidal geometry selected?

The following paragraph about the detector geometry has been inserted in section 2.1.

“The design of GRIT is based on a conical-shaped set of 8 trapezoidal telescopes in both the forward and backward hemisphere with respect to the beam direction assembled with a ring of squared-shape silicon telescopes around 90°, achieving ∼4π solid angle.” …”from which we have selected one with trapezoidal geometry since the length of the strips changes across the detector active area”

2. How should formula (3) be interpreted? If the component of equation Wα is the sum of the individual components derived from the various parameters such as WS (a), Wp (b) and Wc (c), then when this sum is squared we should expect a formula like: a2+2ab+2ac+b2+2bc+c2. What is the mathematical justification for the formula (3) used?

The Wα is not a polynomial, sorry for misleading you. The total contribution of a particle will also be an addition in quadrature. We have modified the formula to avoid confusion. 

3. In the text of the manuscript, there should be references to each formula number.

    References added.

4. The authors mention in several places the quality control of the detector, understood as making sure that the detector units are free of mechanical defects and impurities (dirt, dust, stains, etc.). Assuming that the test system met the highest quality requirements, the question arises as to what specific contaminants the authors write about in their conclusions…

The “contaminants/impurities” that we mention are within the bulk material, this is stated in the conclusions section. We have also added a new sentence saying that further studies will be performed to get more details about it.

Reviewer 4 Report

I recommend the publication of the manuscript after a minor revision.

1. Before Line 1: minor mistake – Insert one sentence, such as *Correspondence: J.A. Dueñas, e-mail: ................

2. Line 26: Re-formulate the sentence:” See for example [6–9].” Also, the invitations, such as: “see.....”

3. Line 48” minor mistake: “....providing the the best...”

4. Line 55: explain why you choose for comparison only this manufacturer “Micron Semiconductor Ltd. [10] ....”. Are these ones the best choice in the industry and why?

5. The title of Table 1 must contain the name of the manufacturer and a reference to it.

6. Line 59-61. Specify all technical limitations of this device, according to the proposed reference.

7. Lines 65-66. Explain the technical/economical reasons applied for choosing this “Test bench block diagram”. Is this solution the optimal one? Why?  Insert a comparison with other Test bench block diagrams used in the industry.

8. Lines 72-73. Give more details about “...to reduce the tension that can build up...”. Is it a technical method applied in the industry? Could you specify any mathematical formula or specify a range of  the reduction of the tension?

9. Can you specify the flexibility in test bench technology correlated with the product solutions?

10. The title of the Table 2 must contain the name of the manufacturer and a reference to it.

11. Line 109: can you insert a mathematical formula for the energy spectrum?

12. Line 156: verify again the proposed formula. Are you sure that (epsilon = 11.9 for Si)? Or ε r ?

13. Insert references for all mathematical formulas.

14. Line 158: verify again the proposed formula (there is a combination of units included in the formula). Explain all parameters.

15. Line 189, and so on. Insert all full names associated with abbreviations (such as DUT, DQC...).

16. Line 271: Insert a paragraph with Statistical analyses, and explain the method, the software used, and all the parameters related to these statistical experiments.

17. Can you insert all the mathematical formulas with the corresponding parameters applied for Section 2?

18: Lines 341-349: minor mistake: insert Institutional Review Board Statement, Informed Consent Statement, Data Availability Statement, and so on.

19. Even though the work is relevant to the journal's scope, i.e., Sensors, I do not find even a single article published in the journal in the list of references.

20. Specify the limits of this study.

This paper can be published after the mentioned revisions.

Minor editing of English language required

Author Response

The authors would like to thank the reviewer for the feedback received and also to acknowledge the improvement of the new version of the paper due to the reviewer’s comments. Please, find our reply to every point made:

I recommend the publication of the manuscript after a minor revision.

  1. Before Line 1: minor mistake – Insert one sentence, such as *Correspondence: J.A. Dueñas, e-mail: …………… Done!
  2. Line 26: Re-formulate the sentence:” See for example [6–9].” Also, the invitations, such as: “see…..Done!
  3. Line 48” minor mistake: “....providing the the best…” Done!
  4. Line 55: explain why you choose for comparison only this manufacturer “Micron Semiconductor Ltd. [10] ....”. Are these ones the best choice in the industry and why?

Very few companies will embark on highly challenging custom design such as ours. Therefore, one has very little choice in the industry of semiconductor detectors. Micron has many years of experience and will give a final product (wafer+masks+mechanics+etc) so it was our best choice. For obvious reasons we cannot make publicity of Micron beyond the technical data of the detectors.

  1. The title of Table 1 must contain the name of the manufacturer and a reference to it...... Done!
  2. Line 59-61. Specify all technical limitations of this device, according to the proposed reference. 

A new sentence with the technical limitations has been added.

  1. Lines 65-66. Explain the technical/economical reasons applied for choosing this “Test bench block diagram”. Is this solution the optimal one? Why?  Insert a comparison with other Test bench block diagrams used in the industry. 

A new sentence explaining the technical/economical reasons has been added at the beginning of section 2. Test benches are very particular and mostly “homemade” so comparisons between different approaches cannot industrially be done.  

  1. Lines 72-73. Give more details about “...to reduce the tension that can build up...”. Is it a technical method applied in the industry? Could you specify any mathematical formula or specify a range of  the reduction of the tension?

Unfortunately, we cannot give any number of mathematical formulas for the tension. We have applied our working experience with these kinds of kapton flex cables.

  1. Can you specify the flexibility in test bench technology correlated with the product solutions?

One may find a modular test bench on the market for standard single pad detectors, or even just for strip detectors with a few number of strips. However, as mentioned in point 7, in our case the singularity of our detectors (i.e. shape, # strips, requirements, etc.) makes impossible to match our need with existing test bench technology, at least not as a whole.

  1. The title of the Table 2 must contain the name of the manufacturer and a reference to it....... Done!
  2. Line 109: can you insert a mathematical formula for the energy spectrum?

We are not sure what the reviewer means. The energy resolution is calculated as the FWHM of the peak distribution from the histogram of the signal amplitudes (spectrum). This is explained in section 2.4 as the fourth and last stage of the quality control.

  1. Line 156: verify again the proposed formula. Are you sure that (epsilon = 11.9 for Si)? Or ε r ?..... Formula modified to help readers 
  2. Insert references for all mathematical formulas.... References added.
  3. Line 158: verify again the proposed formula (there is a combination of units included in the formula). Explain all parameters..... Done!
  4. Line 189, and so on. Insert all full names associated with abbreviations (such as DUT, DQC…).... Full names associated with abbreviations DUT and DQC are given previously.
  5. Line 271: Insert a paragraph with Statistical analyses, and explain the method, the software used, and all the parameters related to these statistical experiments.

There is not really much of a statistical analysis in reference to table 3. As explained before in section 2.3, once the signal has been digitized the ADC software controller (CoMPASS) will apply basic algorithms such the gaussian fit and trapezoidal filtering. The measured energy of each particle (for every strip) is then passed as a numerical value in a file.

  1. Can you insert all the mathematical formulas with the corresponding parameters applied for Section 2?... All parameters values needed are now given in section 2.

18: Lines 341-349: minor mistake: insert Institutional Review Board Statement, Informed Consent Statement, Data Availability Statement, and so on.... Done!

  1. Even though the work is relevant to the journal's scope, i.e., Sensors, I do not find even a single article published in the journal in the list of references.

I’m afraid I cannot comment on that. We hope this special issue will help with this.

  1. Specify the limits of this study.

This study is limited to: (i) Si strip detectors, (ii) number of electronic channels no more than 256, (iii) alpha particle source and  (iv) electronic limitations e.g. equipment noise, inputs, resolutions, etc. We believe these are mentioned throughout the text.

Round 2

Reviewer 2 Report

While not all objections have been addressed satisfactorily, the changes made to the manuscript represent an improvement. I have no object to the the publication at this stage, but I still strongly suggest the following changes:

1) add one of two drawings showing the internal structure of the detector (doped regions, geometry etc). Photos can not provide a clear understanding of the inner operation of the device.

2) Just remove the reference to a Gaussian distribution, unless there are specific reasons to belive that such a distribution should be obtained. I do not see how a clarification on the way in which the source is mounted should justify such a statement. 

3) It seems to me that the stamement "The energy-triangle may be a tool to assess the uniformity of the interstrip SiO2 material and also it can pinpoint possible electric field anomalies at the edges of the strips." should be replaced with something like "We believe that it is worth investigating on the possibility that the enargy-tiangle, as defined in this paper, may be used as a tool for assessing the quality of the interstrip structure". Indeed, if I am not mistaken, at present there is not evidence that the "energy-triangle" can be useful in any way to assess the quality and reliability of the detector. 

Author Response

1) add one of two drawings showing the internal structure of the detector (doped regions, geometry etc). Photos can not provide a clear understanding of the inner operation of the device. Done, see Fig. 9.

2) Just remove the reference to a Gaussian distribution, unless there are specific reasons to belive that such a distribution should be obtained. I do not see how a clarification on the way in which the source is mounted should justify such a statement. Reference to a Gaussian distribution removed.

3) It seems to me that the stamement "The energy-triangle may be a tool to assess the uniformity of the interstrip SiO2 material and also it can pinpoint possible electric field anomalies at the edges of the strips." should be replaced with something like "We believe that it is worth investigating on the possibility that the enargy-tiangle, as defined in this paper, may be used as a tool for assessing the quality of the interstrip structure". Indeed, if I am not mistaken, at present there is not evidence that the "energy-triangle" can be useful in any way to assess the quality and reliability of the detector. Sentence modified following the reviewer's advice.